# Influence of Dietary n-3 Long Chain Polyunsaturated Fatty Acid Intake on Oxylipins in Erythrocytes of Women with Rheumatoid Arthritis

**DOI:** 10.3390/molecules28020717

**Published:** 2023-01-11

**Authors:** Helen M. Lindqvist, Anna Winkvist, Inger Gjertsson, Philip C. Calder, Aaron M. Armando, Oswald Quehenberger, Roxana Coras, Monica Guma

**Affiliations:** 1Department of Internal Medicine and Clinical Nutrition, Institute of Medicine, Sahlgrenska Academy, University of Gothenburg, 405 30 Gothenburg, Sweden; 2Department of Rheumatology and Inflammation Research, Institute of Medicine, Sahlgrenska Academy, University of Gothenburg, 405 30 Gothenburg, Sweden; 3School of Human Development and Health, Faculty of Medicine, University of Southampton, Southampton SO16 6YD, UK; 4NIHR Southampton Biomedical Research Centre, University Hospital Southampton NHS Foundation Trust and University of Southampton, Southampton SO16 6YD, UK; 5Department of Pharmacology, School of Medicine, University of California San Diego, 9500 Gilman Drive, La Jolla, CA 92093, USA; 6Department of Medicine, School of Medicine, University of California San Diego, 9500 Gilman Drive, La Jolla, CA 92093, USA

**Keywords:** rheumatoid arthritis, seafood, n-3 LC PUFA, fatty acids, oxylipins, erythrocytes, eicosanoids

## Abstract

Oxylipins derived from n-3 fatty acids are suggested as the link between these fatty acids and reduced inflammation. The aim of the present study was to explore the effect of a randomized controlled cross-over intervention on oxylipin patterns in erythrocytes. Twenty-three women with rheumatoid arthritis completed 2 × 11-weeks exchanging one cooked meal per day, 5 days a week, for a meal including 75 g blue mussels (source for n-3 fatty acids) or 75 g meat. Erythrocyte oxylipins were quantified by liquid chromatography–tandem mass spectrometry (LC–MS/MS). The results were analyzed with multivariate data analysis. Orthogonal projections to latent structures (OPLS) with effect projections and with discriminant analysis were performed to compare the two diets’ effects on oxylipins. Wilcoxon signed rank test was used to test pre and post values for each dietary period as well as post blue-mussel vs. post meat. The blue-mussel diet led to significant changes in a few oxylipins from the precursor fatty acids arachidonic acid and dihomo-ɣ-linolenic acid. Despite significant changes in eicosapentaenoic acid (EPA) and docosahexaenoic acid (DHA) and free EPA in erythrocytes in the mussel group, no concurrent changes in their oxylipins were seen. Further research is needed to study the link between n-3 fatty-acid intake, blood oxylipins, and inflammation.

## 1. Introduction

Oxylipins, including eicosanoids, are a class of bioactive lipids that mediate inflammation, cytokine synthesis, and cell communication [1]. The long chain (LC) n-6 fatty acid arachidonic acid (AA) is one of the main precursors of oxylipins, but the n-3 LC polyunsaturated fatty acids (PUFAs) eicosapentaenoic acid (EPA) and docosahexaenoic acid (DHA) follow the same cyclooxygenase (COX), lipoxygenase (LOX), and cytochrome P450 (CYP) pathways for generation of oxylipins. In general, EPA results in fewer inflammatory prostaglandins, thromboxanes, and leukotrienes than those produced from AA, and EPA and DHA generate inflammation-resolving mediators such as resolvins, protectins, and maresins [2].

The reduced risk for cardiovascular disease associated with intake of n-3 LC PUFAs is explained by the well-documented beneficial effect on blood lipids [2,3,4], but also by the proposed anti-inflammatory effect caused by fewer inflammatory and inflammation resolving oxylipins [2]. Studies have identified changes in oxylipins in plasma and serum after supplementation with EPA and DHA [5,6,7,8,9,10,11,12,13,14]. However, plasma is often regarded as suboptimal to measure lipid mediators, and erythrocytes are thought to have a higher potential to produce bioactive metabolites of the LOX and CYP pathways [15]. Erythrocytes are the most abundant cell in the human body (83%) and generation and/or mobilization of many immunomodulators by erythrocytes would suggest at least an indirect role in immunity and inflammation [16]. Oxylipins have been successfully quantified by liquid chromatography–tandem mass spectrometry (LC–MS/MS) in erythrocytes [15,17]. However, changes in levels of oxylipins in erythrocytes have not been studied after either fatty acid supplementation or dietary interventions.

Shifting focus to whole diets, when EPA and DHA intakes are increased by increased intake of seafood, the major source of EPA and DHA, this often coincides with a lower intake of meat, a major source of AA. These changes may result in a different effect on the overall oxylipin pattern than that seen with simple supplementation with EPA and DHA without a dietary change. In fact, no studies of dietary effects on oxylipin patterns have been reported.

A reduction of inflammation because of improved dietary habits would be most valuable in patients with chronic inflammatory diseases. Patients with rheumatoid arthritis (RA), a chronic inflammatory disease characterized by systemic inflammation and joint damage, report that different foods improve or worsen their disease symptoms [18]. We have previously reported beneficial effects from both a dietary intervention with a Mediterranean-style diet, the ADIRA-trial [19] involving a diet rich in fish and low in red meat, and a dietary intervention with blue mussels—the Mussels In Rheumatoid Arthritis (MIRA)-trial [20]. Intervention diets rich in EPA and DHA increased these fatty acids in both studies, and the fatty acid pattern changed significantly in erythrocytes and plasma. Despite this, the effects on inflammation markers were limited [21,22]. Because changes in pro and anti-inflammatory lipid mediators, such as oxylipins, precede changes in other inflammatory markers such as cytokines, it would be of value to establish if oxylipins changed in line with changed dietary fatty-acid intake. 

The aim of this study was to explore oxylipin patterns and single oxylipins in erythrocytes, and the effect of dietary intervention on these among participants in the MIRA-trial. 

## 2. Results

### 2.1. Results from Analysis of Oxylipins and Free Fatty Acids in Erythrocytes

In total, 45 oxylipins were detected in erythrocytes; in addition, four free (f) fatty acids (EPA, DHA, AA, and adrenic acid) in erythrocytes were included in the analysis. Pre and post samples from the two periods from 23 women between 25 and 65 years of age, with established RA and active disease, resulted in 92 analyzed samples, but only nine oxylipins were detected in all samples. Oxylipins were only included in data analysis if the analyte exceeded the lower limit of quantification (LLOQ) in ≥50% of the samples, which was the case for 37 oxylipins. Thus, data for leukotriene B_4_ (LTB_4_), 7-hydroxydocosahexaenoic acid (7-HDoHE), 8-hydroxyeicosatrienoic acid (8-HETrE), 11-HDoHE, 15-oxo-eicosatetraenoic acid (15-oxoETE), 13-oxo-octadecadienoic acid (13-oxoODE), 20-hydroxyeicosatetraenoic acid (20-HETE), and 8(9)-epoxy-eicosatrienoic acid (8(9)-EET) were excluded from statistical analysis. Samples with a concentration below LLOQ were set to LLOQ/2. Precursor fatty acids and oxylipins included in statistical analysis are shown in Figure 1. The oxylipin pattern in baseline samples was not significantly influenced by any factors tested including age, body mass index, disease activity, erythrocyte sedimentation rate, C-reactive protein, or triglycerides (results not shown).

### 2.2. Correlations between Erythrocyte Oxylipins and Free Fatty Acids in Baseline Samples

Figure 2 shows a principal component analysis of erythrocyte oxylipins and free fatty acids. Clusters in the figure were also confirmed in Spearman correlation analysis (Appendix A). Concentrations of all free fatty acids had a strong (r > 0.8, *p* < 0.01) correlation, and for free AA (fAA), free EPA (fEPA), and free DHA (fDHA), a very strong (r > 0.9, *p* < 0.01) correlation in baseline samples, indicating that the concentration of free fatty acids within erythrocytes might be commonly regulated. fEPA had only weak correlation to its oxylipins, whereas fAA and fDHA had moderate (r > 0.4, *p* < 0.05) correlations to some of their oxylipins. The only AA-derived oxylipin that had no correlation to fAA at all was tetranor PGFM. fEPA in erythrocytes had a moderate correlation to total EPA in erythrocytes, at baseline. Finally, there were no correlations between fAA and fDHA and the total content of these fatty acids in erythrocytes.

### 2.3. Correlations between Precursor Fatty Acids and Oxylipins in Erythrocytes at Baseline

Total concentrations of precursor fatty acids in erythrocytes had limited correlation to oxylipins (Appendix A). DHA correlated moderately with 14-HDoHE. EPA correlated moderately with 9- and 11-HEPE. AA correlated moderately with 11(12)-EET, 11,12-diHETrE, 14(15)-EET, and 14,15-diHETrE. No correlations were found between concentrations (mg/mL) or percentage of dihomo-ɣ-linolenic acid, linoleic acid, and α-linolenic acid and concentrations of their oxylipins in erythrocytes at baseline. 

### 2.4. Influence of the Dietary Intervention on Free Fatty Acids in Erythrocytes

fAA did not change during with either mussel or control intervention, despite a decrease in erythrocyte AA during the mussel intervention period (*p* = 0.039). In addition, fEPA and fDHA decreased during the control period (*p* = 0.048, *p* = 0.048), but did not increase during the mussel intervention. fEPA was higher after the mussel intervention than after the control period (*p* = 0.048), which is consistent with erythrocyte total EPA (Table 1). Thus, dietary intake of EPA from blue mussels seems to influence the concentration of fEPA in erythrocytes.

### 2.5. Effect of the Intervention and Control Diets on Oxylipin Patterns and Single Oxylipins

We have previously presented findings showing that patterns of erythrocyte fatty acids could predict all individuals’ dietary periods (mussel intervention vs. control diet) correctly in an OPLS-EP model [21], but this was not possible for oxylipins. Neither OPLS-EP nor OPLS-DA resulted in models of high quality with a good predictability (Table 2). Because of this, selected class discriminating oxylipins from OPLS-EP and OPLS-DA must be evaluated with caution. However, two oxylipins from AA overlapped between the models, the proinflammatory oxylipins 12-HHTrE and 9-HETE. In addition, the anti-inflammatory 15-HETrE was one of the oxylipins from the OPLS-EP model that contributed most to differentiation between the two dietary periods.

There were only three individual oxylipins that changed significantly during the dietary periods. The control diet led to a decrease in the anti-inflammatory 9-HOTrE (*p* = 0.029), an oxylipin produced from α-linolenic acid, which cannot be explained by changes in its precursor fatty acid. The intervention period led to a decrease in the anti-inflammatory oxylipins 5,6-diHETrE and 15-HETrE produced from AA and dihomo-ɣ-linolenic acid, respectively (*p* = 0.012, *p* = 0.024) (Table 1). This could possibly be explained by a decrease in both of their precursor fatty acids. None of the oxylipins differed significantly when comparing concentrations after the two dietary periods (Table 1) or between periods when analyzed with ANCOVA (results not shown). 

## 3. Discussion

In the present study we show, for the first time, erythrocyte oxylipin patterns in patients with RA undergoing a dietary intervention. This study has several strengths including the cross-over design, which is optimal for dietary interventions in patients who have not only different dietary habits to one another, but also variations in disease activity, co-morbidities, and pharmacological treatment. Data on erythrocyte fatty acids both pre and post intervention and control period also improve the possibility of correctly understanding the effects of diet intervention on oxylipins. Study limitations are the small sample size and a limited change in dietary PUFAs that only led to changes in a small number of erythrocyte oxylipins. 

Due to the lack of published results on effects of fatty-acid intake on erythrocyte oxylipin levels, we here discuss our results compared with previous findings in plasma by others. It must be emphasized that oxylipins in erythrocytes and in plasma are not necessarily reflective of each other; the bioactivity of compounds in plasma is often disputed and this was also a reason for our choice of quantifying oxylipins in erythrocytes. In addition, the patients in our study were women with RA and on anti-rheumatic pharmacological treatment and it is not known if the disease itself or anti-rheumatic treatment could influence oxylipin concentrations.

Although no dietary interventions have reported effects on oxylipins, some studies have presented data on changes in plasma or serum oxylipins during EPA and DHA supplementation [5,6,7,8,9,10,11,12,13,14], but the results on specific oxylipins are inconsistent. For most oxylipins reported in this paper, unchanged concentrations of these in plasma have been reported by at least one or two studies with EPA and DHA supplementation, while the same number of studies have, instead, reported changes. Exceptions include only 12,13 diHOME, 9-HODE, and 13-HODE that did not change in any of the studies that reported these oxylipins. This was also expected since the precursor LA did not change in these studies [5,6,7,8,9]. Only three erythrocyte oxylipins were significantly changed during the MIRA-trial, 5,6-DiHETrE, 15-HETrE, and 9-HOTrE. 5,6-DiHETrE has been found to be elevated after intake of non-steroidal anti-inflammatory drugs, suggesting a potential role in anti-inflammatory modulation [24]. 15-HETrE has an antiproliferative effect [25] and can inhibit the biosynthesis of proinflammatory eicosanoids LTB_4_ and 12-HETE [26]. There is little detail about the function of 9-HOTrE, but it is thought that the effect is similar to 13-HOTrE which has multifactorial anti-inflammatory effects [27]. Our results showed that 5,6 diHETrE decreased during the intervention and which also was reported in plasma by Lundstrom et al. [6], but not by Keenan et al. [7], Watkins et al. [10], or Ostermann et al. [5]. 15-HETrE decreased in erythrocytes during the MIRA intervention and similar changes were reported in plasma by Keenan et al. [7] and Shearer et al. [13], but not by Lundstrom et al. [6] or Ostermann et al. [5]. Finally, it was not possible to compare the decrease in 9-HOTrE in erythrocytes during the MIRA control period to that of other studies since these lacked control groups [5,7,8,9]. In contrast to our results, Ostermann et al. [5] reported increases in plasma oxylipins from DHA and 9-HEPE, 11-HEPE, and 18-HEPE from EPA in a dose-dependent way. There are no other studies reporting findings on oxylipins 12-HHTrE, 5,15-diHETE, and tetranor PGFM from AA, even in plasma. 

It is possible that a more pronounced change after increased intake is found for CYP-derived epoxides, but these were not detected in erythrocytes in our study, where only hydroxy fatty acids derived from EPA were successfully quantified. In line with our results, CYP-derived epoxides in erythrocytes were not reported in the study by Fu et al. [17] and were reported to have very low concentrations in the study by Liu et al. [15].

Unexpectedly, mainly AA and DHA oxylipins in erythrocytes contributed to separate the blue mussel and control diets in the multivariate models. This could probably be explained by the broad changes in the fatty-acid pattern in erythrocytes caused by the dietary interventions. Further, the participants in the MIRA trial had a high intake of fish and shellfish in their habitual diet and hence only a moderately increased intake of EPA and DHA during intervention. The lack of effect on the oxylipin pattern could thus be due to an insufficient change in total intake of precursor fatty acids. In fact, Ostermann et al. found that low-dose supplementation of EPA and DHA did not result in changes in oxylipins [5]. In addition, correlations between habitual EPA and DHA intake and EPA and DHA oxylipins in plasma or erythrocytes have not been confirmed; only changes after high dose supplementation have been confirmed [5]. Indeed, only moderate correlations among precursors, free fatty acids, and oxylipins were seen at baseline in the MIRA samples. This indicates that the relationships among total dietary intake of fatty acids and free fatty acids and oxylipin patterns in erythrocytes need further investigation in order to be more fully understood. 

## 4. Materials and Methods

### 4.1. Patients and Experimental Design

The MIRA-trial design and participants have been described in detail previously [20]. In short, 39 women between 25 and 65 years of age, with established RA and active disease (disease activity score 28 joints−erythrocyte sediment rate (DAS28-ESR) > 3.0) were included in the study. To start with they were allocated either the intervention diet (i.e., blue mussels) or the control diet (i.e., meat) in a randomized, single-blinded cross-over intervention with 11 weeks of each diet and an 8-week washout period. The participants continued with their anti-rheumatic pharmacological treatment during the study period.

During the dietary periods participants were provided with five ready-made meals/week (the same thirteen different dishes in both periods). To these meals the participants added either thawed frozen pre-cooked blue mussels (75 g) or the same amount of thawed frozen pre-cooked chicken or, once a week, meatballs. Except for limitations in intake of fish and shellfish, participants otherwise consumed their habitual diets during the whole study period. The blue mussels consumed every week provided 0.24 g saturated fat, 0.14 g monounsaturated fat, 0.47 g polyunsaturated fat, 0.18 g EPA, and 0.16 g DHA daily. The meat provided 0.40 g saturated fat, 0.50 g monounsaturated fat, 0.22 g polyunsaturated fat, 0.00 g EPA, and 0.01 g DHA daily [20].

Compliance was evaluated by self-reporting of the number of dishes consumed and 24 h dietary recalls conducted by telephone midway through each dietary period as well as with plasma and erythrocyte fatty acids. Fasting blood samples were collected before and after each dietary period and were analyzed for free fatty acids and oxylipins in erythrocytes. Erythrocyte fatty acid concentrations have been reported previously [21]. Figure 3 describes the relationship between the presented lipids (total fatty acids, free fatty acids, and oxylipins) in erythrocytes in this paper and their products. Patients’ baseline characteristics have been presented in detail elsewhere [20]. In short, twenty-three women with a median (range) age of 55 (32–66) years, body mass index of 25 (19–37) kg/m^2^, C-reactive protein 2 (0–14) mg/dL, and DAS28-ESR 3.9 (3.1–5.3) completed both dietary periods. Most of the participants were being treated with non-steroidal anti-inflammatory drugs (65%) and disease-modifying anti-rheumatic drugs (61%).

All procedures were conducted according to the Declaration of Helsinki and were approved by Gothenburg Regional Ethical Review Board (25 May 2015/Dnr 230-15 with addendum T879-17 to send samples abroad for analysis), and all participants provided written informed consent. The trial was registered at https://register.clinicaltrials.gov (accessed on 25 May 2015) as NCT02522052.

### 4.2. Analysis of Oxylipins

#### 4.2.1. Sample Collection and Handling

Fasting blood samples were collected into 6 mL lithium heparin vacutainer tubes (BD Vacutainer^®^, Heparin tubes, Franklin Lakes, NJ, USA). Tubes were centrifuged within three hours (room temperature, 913× *g* for 10 min). Erythrocytes were then washed with PBS-sterile liquid (Avantor (VWR life science), Spånga, Sweden) and centrifuged twice (350× *g* for 10 min, room temperature, low brake). All plasma and erythrocyte samples were stored in a −80 °C freezer until analysis. 

#### 4.2.2. Lipid Extraction and LC–MS Measurement of Oxylipins

All samples were stored at −80 °C, thawed once, and immediately used for free fatty acid and oxylipin isolation as previously described [28]. Briefly, 100 μL sample was first sonicated to disrupt the erythrocyte membrane followed by spiking with a cocktail of 26 deuterated internal standards that also included some selected PUFAs (individually purchased from Cayman Chemicals, Ann Arbor, MI, USA) and brought to a volume of 1 mL with 10% methanol. The samples were then purified by solid-phase extraction on Strata-X columns (Phenomenex, Torrance, CA, USA), by an activation procedure consisting of consecutive washes with 3 mL of 100% methanol followed by 3 mL of water. The oxylipins were then eluted with 1 mL of 100% methanol, which was dried under a vacuum, dissolved in 50 μL of buffer A (consisting of water–acetonitrile–acetic acid, 60:40:0.02 [*v*/*v*/*v*]), and immediately used for analysis. Oxylipins in erythrocytes were analyzed and quantified by LC–MS/MS, as previously described [28,29]. Briefly, oxylipins were separated by reverse-phase chromatography with a 1.7 μm 2.1 × 100 mm BEH Shield Column (Waters, Milford, MA, USA) and an Acquity UPLC system (Waters). The column was equilibrated with buffer A, and 10 μL of sample was injected via the autosampler. Samples were eluted with a step gradient starting with 100% buffer A for 1 min, then to 50% buffer B (consisting of 50% acetonitrile, 50% isopropanol, and 0.02% acetic acid) over a period of 3 min, and then to 100% buffer B over a period of 1 min. The LC was interfaced with an IonDrive Turbo V ion source, and mass spectral analysis was performed on a triple quadrupole AB SCIEX 6500 QTrap mass spectrometer (AB SCIEX, Framingham, MA, USA). Oxylipins were measured by electrospray ionization in negative-ion mode and multiple reaction monitoring (MRM) by the most abundant and specific precursor ion/product ion transitions to build an acquisition method capable of detecting 158 analytes and 26 internal standards. The ion spray voltage was set at −4500 V at a temperature of 550 °C. Collisional activation of the oxylipin precursor ions was achieved with nitrogen as the collision gas with the declustering potential, entrance potential, and collision energy optimized for each metabolite. Oxylipins were identified by matching their MRM signal and chromatographic retention time with those of pure identical standards.

Oxylipins were quantified by the stable isotope dilution method. Briefly, identical amounts of deuterated internal standards were added to each sample and to all the primary standards used to generate standard curves. To calculate the amount of oxylipins and free fatty acids in a sample, ratios of peak areas between endogenous metabolite and matching deuterated internal standards were calculated. Ratios were converted to absolute amounts by linear regression analysis of standard curves generated under identical conditions. Oxylipin levels are expressed in picomole/milliliter (pmol/mL). To account for batch effects, quality control samples were run in each batch; the average coefficient of variance for the quantified oxylipins was 4% (standard deviation 0.01).

### 4.3. Statistical Methods

#### 4.3.1. Pre-Processing of Data

Relative changes in total erythrocyte AA, EPA, and DHA were calculated from the individual data or the group means against baseline with the formula: relative change (%) = conc(t)/conc(t0) × 100.

#### 4.3.2. Multivariate Methods

All multivariate analyses were done in SIMCA software v.17.0 (Sartorius Stedim Data Analytics AB, Umeå, Sweden) and no samples were excluded from any of the analyses. Principal component analysis (PCA) models and orthogonal projections to latent structures (OPLS) were used to explore clustering patterns of observations and trends in the data in relation to known factors and outliers (*n* = 23). OPLS models included not only x-variables such as oxylipin data but also y-variables, i.e., additional known factors that could influence the data, such as DAS28-ESR, body mass index, triglycerides, and age. Separation of classes and variables related to separation in the data according to classification of diet (intervention vs. control diet) were evaluated with an orthogonal projections to latent structures with effect projections (OPLS-EP), where delta values between periods (post mussels − post meat) were used since the samples were paired. In addition, OPLS with discriminant analysis (OPLS-DA), where delta values for each period (post-pre) were used (*n* = 46), was performed. Cross-validation groups were set to 23 (equal to the number of study participants) in OPLS-DA and were based on individual ID. The validity of OPLS-DA models was assessed by permutation tests (*n* = 999). In addition, cross-validated predictive residuals (CV-ANOVA), the cumulative amount of explained variation in the data summarized by the model (R2X [cum] and R2Y [cum]), and the predictive ability of the model (Q2 [cum]) are presented. 

Class discriminating oxylipins of interest from the OPLS-EP and OPLS-DA models were selected if loadings w ≥ ±0.1 and if they were among the twenty highest variable importance (VIP) scores.

#### 4.3.3. Univariate Methods

Statistical analyses were done in SPSS version 28 (SPSS Inc., Chicago, IL, USA). Wilcoxon signed rank test was used to test pre and post values for each dietary period as well as post intervention vs. post control values. If any change for an oxylipin was identified, treatment effect of the dietary intervention was evaluated by a mixed-effect analysis of covariance (ANCOVA) model with treatment (intervention or control) and sequence as fixed effects and subject nested in sequence as a random effect. Pre-dietary period values for the fatty acid were included as a covariate.

Spearman’s correlation analysis was performed to evaluate associations between oxylipins, free fatty acids, and total fatty acids in baseline samples (*n* = 23). In this explorative study, results are presented as mean (s.d.) with significance set at α = 0.05, i.e., correction for multi-testing was not performed.

## 5. Conclusions

This well controlled cross-over dietary intervention with significant changes in erythrocyte fatty acids and fEPA did not result in concurrent changes in oxylipins in erythrocytes. Mussel intervention led to changes in a few oxylipins from the precursor fatty acids AA and dihomo-ɣ-linolenic acid, indicating that dietary changes may have a wider but less pronounced effect on oxylipins than high dose n-3 PUFA supplementation. Further research is needed to study the link between n-3 fatty-acid intake, blood oxylipins, and inflammation.

## Figures and Tables

**Figure 1 molecules-28-00717-f001:**
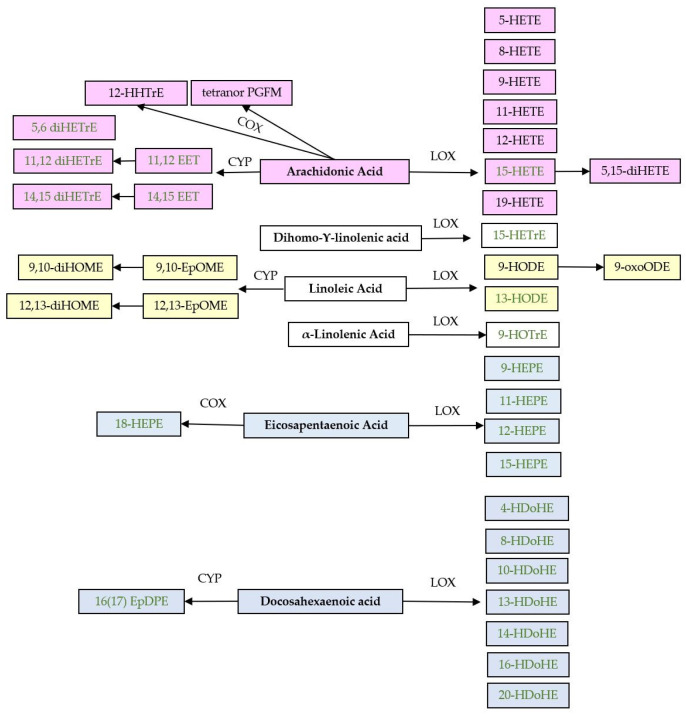
Precursor fatty acids and oxylipins. Enzymes: cyclooxygenase (COX), lipoxygenase (LOX), and cytochrome P450 (CYP). Oxylipins: prostaglandin F Metabolite (PGFM), hydroxyheptadecatrienoic acid (HHTre), hydroxyoctadecatrienoic acid (HOTrE), epoxyoctadecenoic acid (EpOME), dihydroxy-octadecenoic acid (diHOME), hydroxyoctadecadienoic acid (HODE), hydroxyeicosapentaenoic acid (HEPE), hydroxydocosahexaenoic acid (HDoHE), hydroxyeicosatrienoic acid (HETrE), dihydroxyeicosatrienoic acid (di-HETrE), oxo-octadecadienoic acid (oxoODE), hydroxyeicosatetraenoic (HETE), dihydroxyeicosatetraenoic acid (diHETE), epoxy-eicosatrienoic acid (EET), and epoxy-docosapentaenoic acid (EpDPE). Green text indicates that the oxylipin has anti-inflammatory effects [23].

**Figure 2 molecules-28-00717-f002:**
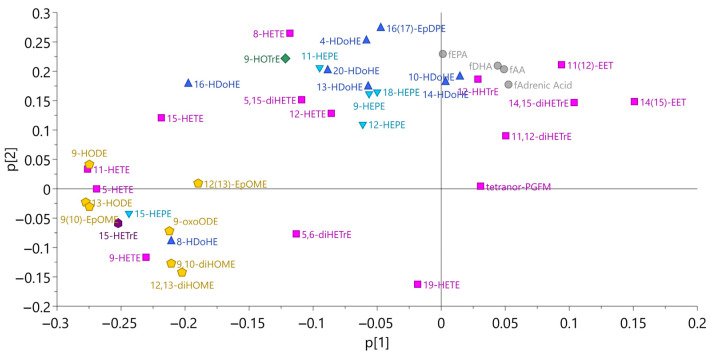
Principal component analysis for oxylipins and free fatty acids in erythrocytes (*n* = 23, model 1). The same color and shape mean that they belong to the same precursor fatty acid. Pink boxes, arachidonic acid; yellow pentagon, linoleic acid; blue triangle, docosahexaenoic acid; inverted blue triangle, eicosapentaenoic acid; green diamond, α linolenic acid; and purple hexagon, dihomo-ɣ-linolenic acid.

**Figure 3 molecules-28-00717-f003:**
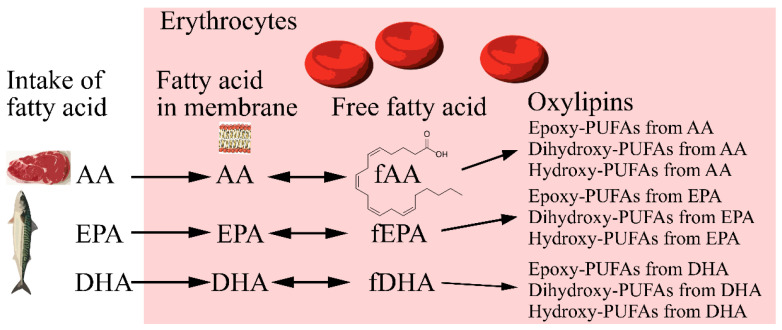
Relationship between dietary fatty acids, fatty acids in erythrocytes, and oxylipins. AA, arachidonic acid; EPA, eicosapentaenoic acid; DHA, docosahexaenoic acid; f free, PUFA poly unsaturated fatty acid.

**Table 1 molecules-28-00717-t001:** Fatty acids, free fatty acids, and oxylipins with significant changes in erythrocytes before and after blue mussel and control diet periods.

	Control			Blue Mussel			
Free Fatty Acids/Oxylipin	Pre	Post	P (Pre vs. Post)	Pre	Post	P (Pre vs. Post)	P (Post BM vs. Post C)
**α-linolenic acid (18:3 n-3) ^a^**	1.26 (0.98,1.74)	1.35 (0.95,1.61)	0.738	1.33 (1.19,1.6)	1.37 (1.08,1.73)	0.563	0.761
*Free α-linolenic acid (18:3 n-3) ^b^*	ND	ND		ND	ND		
9-HOTrE ^b^	0.72 (0.48,1.39)	0.49 (0.28,0.72)	**0.029**	0.46 (0.30,0.77)	0.42 (0.31,0.72)	0.465	0.758
**Dihomo-** **ɣ-linolenic acid (20:3 n-6) ^b^**	12.6 (10.2,14.6)	13.5 (10.6,15.9)	**0.036**	13.2 (11.5,15.4)	12.2 (10.1,14.8)	**0.002**	0.073
*Free dihomo-* *ɣ-linolenic acid (20:3 n-6) ^b^*	ND	ND		ND	ND		
15-HETrE ^b^	2.26 (1.40,3.15)	2.26 (1.04,3.29)	0.670	2.11 (1.34,3.14)	1.87 (1.24,2.75)	**0.024**	0.171
**Arachidonic acid (20:4 n-6) ^a^**	108.6 (91.0,125.1)	112.8 (98.2,125.3)	0.128	114.8 (105.9,128.6)	111.0 (100.2,116.5)	**0.039**	0.465
*Free arachidonic acid (20:4 n-6) ^b^*	9820 (1650,13770)	2890 (1660,7500)	0.144	3940 (1550,7840)	4280 (1300,7650)	0.627	0.627
5,6-diHETrE ^b^	1.12 (0.68,1.87)	0.57 (0.08,1.62)	0.201	0.90 (0.09,1.90)	0.36 (0.09,0.87)	**0.012**	0.108
**Adrenic acid (22:4 n-6)**							
*Free adrenic acid (22:4 n-6) ^b^*	3390 (1870,6650)	1730 (1060,3380)	0.171	2080 (1250,4900)	2200 (770,3210)	0.465	0.761
**Eicosapentaenoic acid (20:5 n-3) ^a^**	10.49 (8.09,12.53)	8.78 (6.86,10.82)	0.212	10.51 (7.54,12.22)	12.24 (10.27,14.46)	**0.001**	**<0.001**
*Free eicosapentaenoic acid (20:5 n-3) ^b^*	321 (99,424)	125 (87,194)	**0.048**	179 (97,270)	207 (113,330)	0.967	**0.048**
**Docosahexaenoic acid (22:6 n-3) ^a^**	47.6 (39.3,53.4)	45.0 (39.6,48.9)	0.627	47.6 (40.4,56.4)	47.0 (42.9,53.1)	0.584	**0.007**
*Free docosahexaenoic acid (22:6 n-3) ^b^*	2500 (481,4160)	628 (370,1880)	**0.048**	1140 (298,2210)	1120 (350,2110)	0.761	0.236

^a^ mg/mL, ^b^ pmol/mL, Wilcoxon signed rank test was used for all statistics. Values are median (IQR). ND, not detected. BM = blue mussel, C = control.

**Table 2 molecules-28-00717-t002:** Model statistics for oxylipins in erythrocytes.

Model	Scaling	Nr of Lv ^a^	N	R2X [cum] ^b^	R2Y [cum] ^c^	Q2 [cum] ^d^	CV-ANOVA ^e^ (*p*-Value)	Permutation Test (Q2) ^f^	Correct Classified (%C/%M)
PCA	UV	4	23	0.694		0.063			
OPLS-EP mussels vs. control	UVN	1 + 1 + 0	23	0.629	0.974	−0.0268			
OPLS-DA Δmussels vs. Δcontrol ^g^	UV	1 + 1 + 0	46	0.588	0.208	−0.142	1	−0.132 ^h^	48/96

^a^ Latent variables; ^b^ cumulative fraction of the sum of squares of X explained by the selected latent variables; ^c^ cumulative fraction of the sum of squares of Y explained by the selected latent variables; ^d^ cumulative fraction of the sum of squares of Y predicted by the selected latent variables, estimated by cross validation; ^e^ analysis of variance testing of cross-validated predictive residuals; ^f^ the intercept between real and random models, degree of overfit; ^g^ model with forced two first components; ^h^ permutation regarded as not acceptable %C = percentage of control diet samples; %M = percentage of blue mussel diet samples.

## Data Availability

Data cannot be shared publicity because of Swedish law. The datasets analyzed in the current study are available from the corresponding author Helen Lindqvist on reasonable request.

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
