# Peer review of "Influence of Dietary n-3 Long Chain Polyunsaturated Fatty Acid Intake on Oxylipins in Erythrocytes of Women with Rheumatoid Arthritis"

_molecules, 2023, doi:10.3390/molecules28020717_

Round 1
Reviewer 1 Report
In the current manuscript the authors present the result of the analysis of oxylipins in erythrocytes from a cohort of rheumatoid arthritis patients subjected to two different diet supplements, namely mussels and chicken, for 11 weeks plus an 8-week washout. The results indicate modest differences in the presence of three oxylipins (9-HOTrE, 15-HETrE, 5,6-diHETrE) and their precursor fatty acids (alha-linolenic, di-homo-gamma linolenic, arachidonic) as well as EPA and DHA, between both diets and before and after intervention.
Comments:
- Although the general question behind the study is very interesting, that of the impact of diet on oxylipin production, and the fact that this is a well-controlled cohort, the study presents a number of limitations, mainly two: the study is limited to one cell type and compartment, it is limited to RA patients (no healthy controls). Other limitations are commented below. Altogether, all these limitations also limit the relevance of the results.
- As said in lines 184-185, the disease and the treatment could influence the results. Ideally, the patients should be stratified by disease severity. Or, alternatively, the correlation between oxylipin levels and the disease activity score could be evaluated; or a correlation between oxylipin levels and the evolution of the score between the beginning and the end of the intervention (delta).
- Patient information should be presented before the results are presented, either within the results or by placing the materials and methods section right after the introduction. Maybe just a reminder would be enough, by indicating the number of patients per group. Even if the details of the cohort are given in a separate publication, a minimum is necessary to follow the reported data.
- The composition in fatty acids of the complement diets should be reported (ideally the contents in the total diet, although this is probably too complicated). As said in the discussion, the diet complement could be too moderate to expect an influence in the oxylipin profiles.
- The anti-rheumatic pharmacological treatment can also exert an impact on the parameters measured. This is another limitation the study, and this is why ideally these data should be compared with similar results obtained from an age/sex matched cohort of healthy individuals.
- The rationale of the study, as it is written (lines 77-80), is hard to understand: which inflammatory markers were analyzed in the cited studies, and why analyzing oxylipins would be less challenging? Also, oxylipins are presented as anti-inflammatory, while oxylipins could also be pro-inflammatory.
- It is said that “the oxylipin pattern in baseline samples was not significantly influenced by any factors tested including age, body mass index, disease activity, erythrocyte sedimentation rate, C-reactive protein or triglycerides (results not shown)” (lines 93-96. In fact, it would be good to show those results. At least on a supplemental table containing all the quantified oxylipins.
- On lines 109-119 the authors mention some correlations. It would also be good to show all correlations in table or a supplemental table. The same comment applies to the next paragraph (lines 127-133).
- As far as I understand, the table cited on line 165 should be Table 1.
- On Table 1 it should be well defined what “post-post” means.
- What is known about the function of the three differential oxylipins? Nothing is mentioned on the subject.
- It is not clear why meatballs are included in the diet, or if this has any particular relevance.
Author Response
Thank you for your time reviewing our manuscript. Reviewer comments and questions in italic text.
In the current manuscript the authors present the result of the analysis of oxylipins in erythrocytes from a cohort of rheumatoid arthritis patients subjected to two different diet supplements, namely mussels and chicken, for 11 weeks plus an 8-week washout. The results indicate modest differences in the presence of three oxylipins (9-HOTrE, 15-HETrE, 5,6-diHETrE) and their precursor fatty acids (alha-linolenic, di-homo-gamma linolenic, arachidonic) as well as EPA and DHA, between both diets and before and after intervention.
Comments:
- Although the general question behind the study is very interesting, that of the impact of diet on oxylipin production, and the fact that this is a well-controlled cohort, the study presents a number of limitations, mainly two: the study is limited to one cell type and compartment, it is limited to RA patients (no healthy controls). Other limitations are commented below. Altogether, all these limitations also limit the relevance of the results.
Thank you for your valuable comments. We agree that it would have been nice to also present plasma oxylipins, but funding did not cover to analyze both. We also agree that it would be interesting to compare patients with RA and healthy individuals, but this was not the aim of the presented work and will hopefully be investigated further in a larger setting. Our aim was to evaluate if oxylipin changes followed the clinical improvement that was seen in DAS28 in this group of patients.
- As said in lines 184-185, the disease and the treatment could influence the results. Ideally, the patients should be stratified by disease severity. Or, alternatively, the correlation between oxylipin levels and the disease activity score could be evaluated; or a correlation between oxylipin levels and the evolution of the score between the beginning and the end of the intervention (delta).
The number of patients is unfortunately too small for stratification analysis. We have evaluated oxylipins patterns with disease activity in our OPLS analysis of oxylipin patterns, and this showed that oxylipin patterns were not influenced by age, body mass index, disease activity, erythrocyte sedimentation rate, C-reactive protein or triglycerides (row 94-96) at baseline To avoid multi testing, we decided to not do univariate testing if the multivariate models did not capture correlations. In addition, no significant changes in oxylipins were seen between the two diets. If significant changes had been found, we would have adjusted for factors that influenced the results.
- Patient information should be presented before the results are presented, either within the results or by placing the materials and methods section right after the introduction. Maybe just a reminder would be enough, by indicating the number of patients per group. Even if the details of the cohort are given in a separate publication, a minimum is necessary to follow the reported data.
Here we follow the journal instructions placing the material and methods after the results, but we have now also added the number of samples and participants in the first part of the results “Pre and post samples from the two periods from 23 women between25 and 65 years of age, with established RA and active disease resulted in 92 analyzed samples” at row 77-78
- The composition in fatty acids of the complement diets should be reported (ideally the contents in the total diet, although this is probably too complicated). As said in the discussion, the diet complement could be too moderate to expect an influence in the oxylipin profiles.
Because this is a cross-over designed study the individuals are compared to themselves, and the habitual diet should therefore not differ too much. In addition, the dishes were identical in both periods except for the blue mussels and meat (chicken and meatballs) and differences in fatty acid intake should mainly be due to the intervention i.e. from blue mussels vs meat. We have changed the sentence at row 239 to explain this: “…the same thirteen different dishes in both periods” and added:
. The blue mussels consumed every week provided 0.24g saturated fat, 0.14g mono unsaturated fat, 0.47g poly unsaturated fat, 0.18g EPA and 0.16g DHA daily. The meat provided 0.40g saturated fat, 0.50g mono unsaturated fat, 0.22g poly unsaturated fat, 0.00g EPA and 0.01g DHA daily.
ecause it is very hard to capture 11-weeks of dietary intake objectively we have analyzed both erythrocytes and plasma to control for fatty acid intake. Because our data on fatty acids in erythrocytes and plasma showed significant changes between the two dietary periods (despite rather modest intake of n-3 fatty acids), we wanted to find out if also oxylipins were significantly changed.
- The anti-rheumatic pharmacological treatment can also exert an impact on the parameters measured. This is another limitation the study, and this is why ideally these data should be compared with similar results obtained from an age/sex matched cohort of healthy individuals.
Unfortunately, there are very few studies in this field to compare our data with. We also agree that further studies in the field is necessary to understand the influence of diet on oxylipins.
- The rationale of the study, as it is written (lines 77-80), is hard to understand: which inflammatory markers were analyzed in the cited studies, and why analyzing oxylipins would be less challenging? Also, oxylipins are presented as anti-inflammatory, while oxylipins could also be pro-inflammatory.
Thank you for pointing this out. We have tried to make improve our point in this sentence and changed it to: pro- and anti-inflammatory.
- It is said that “the oxylipin pattern in baseline samples was not significantly influenced by any factors tested including age, body mass index, disease activity, erythrocyte sedimentation rate, C-reactive protein or triglycerides (results not shown)” (lines 93-96. In fact, it would be good to show those results. At least on a supplemental table containing all the quantified oxylipins.
We have added data on univariate spearman correlation coefficients in supplemental tables as suggested by the reviewer
- On lines 109-119 the authors mention some correlations. It would also be good to show all correlations in table or a supplemental table. The same comment applies to the next paragraph (lines 127-133).
We have added supplemental table 1 and 2 with correlation coefficients and p-values for spearman correlation coefficients.
- As far as I understand, the table cited on line 165 should be Table 1.
Correct, thank you for noticing this.
- On Table 1 it should be well defined what “post-post” means.
All p-values in Table 1 have now a description explaining if the test is pre values vs. post values or post blue mussel vs. post control. We hope that this makes it easier to understand.
- What is known about the function of the three differential oxylipins? Nothing is mentioned on the subject.
We didn’t want to get too deep into specific mechanisms before other studies have confirmed possible effects from dietary n-3 fatty acids since we only found significant changes within treatment periods and not when comparing intervention and control. We have now added this short section at row 203-209.
15-HETrE has an antiproliferative effect (1) and can inhibit the biosynthesis of proinflammatory eicosanoids leukotriene B4 (LTB4) and 12-hydroxyeicosatetraenoic acid (12-HETE)(2). There is little detail about the function of 9-HOTrE, but it is thought that the effect is similar to 13-HOTrE which has multifactorial anti-inflammatory effects (3). 5,6-DiHETrE has been found to be elevated after intake of non-steroidal anti-inflammatory drugs, suggesting a potential role in anti-inflammatory modulation (4).
- It is not clear why meatballs are included in the diet, or if this has any particular relevance.
For the control diet we wished to compare the dishes with meat with an average fat content similar to the blue mussels and that reflect meat intake without exaggerating it to a directly unhealthy control diet (only red meat). In addition, we needed to serve meals that tasted good with either blue mussels or meat and that were easy to just thaw and add to the meals distributed. The combinations of all these factors resulted in 4 dishes with chicken and 1 with meatballs per week.
Reviewer 2 Report
In this work the authors study the effect of two diets, with different composition in fatty acids, in the relative abundance of oxylipins in erytrocite membranes.
Experimental designs and statistic treatment are adequate, being the conclusions relevant to the field, although a higher n in the dietary intervention study would have been desirable.
Results are well presented, but figure 2 should include probability ellipsoids, as well as a legend in order to add clarity.
Author Response
Thank you for your time reviewing our manuscript. Reviewer comments and questions in italic text.
In this work the authors study the effect of two diets, with different composition in fatty acids, in the relative abundance of oxylipins in erytrocite membranes.
Experimental designs and statistic treatment are adequate, being the conclusions relevant to the field, although a higher n in the dietary intervention study would have been desirable.
We agree, but unfortunately it is very hard to recruit patients to consume blue mussels five days a week for 11 weeks and also difficult to distribute foods to large groups of participants.
Results are well presented, but figure 2 should include probability ellipsoids, as well as a legend in order to add clarity.
We have added a legend to add clarity, but the PCA only includes baseline data and there are no groups since all participants are compared to themselves. It is not clear to us for what groups probability ellipsoids should be added to add information of interest.
Reviewer 3 Report
The authors of this article have taken up a very interesting topic regarding the influence of dietary n-3 long chain polyunsaturated fatty acid 2 intake on oxylipins in erythrocytes in women with rheumatoid 3 arthritis. The work is prepared in a very good way, and the selection of the study group is amazing. A small downside is the small size of the study population (only 23 women). Reading the above article, a few minor remarks come to mind, which I present below:
1. The abstract lacks information why the blue mussel diet was used in the study? It is worth adding a sentence regarding the relationship between erythrocytes and oxylipins and inflammation.
2. And paragraph 2 (results) abbreviations appear LTB4, 7-HDoHE, 8-HETrE, 11-HDoHE, 90 15-oxoETE, 13-oxoODE, 20-HETE, and 8(9)-EET) - it is worth explaining them in the article.
3. There are abbreviations throughout the article that have no explanation (e.g. fAA, fEPA, fDHA). Please re-read the text and explain all the abbreviations used.
4. I understand that the full characteristics of the patients included in the study called MIRA were given in another article - reference number 20.
Author Response
Thank you for your time reviewing our manuscript. Reviewer comments and questions in italic text.
The authors of this article have taken up a very interesting topic regarding the influence of dietary n-3 long chain polyunsaturated fatty acid 2 intake on oxylipins in erythrocytes in women with rheumatoid 3 arthritis. The work is prepared in a very good way, and the selection of the study group is amazing. A small downside is the small size of the study population (only 23 women). Reading the above article, a few minor remarks come to mind, which I present below:
- The abstract lacks information why the blue mussel diet was used in the study? It is worth adding a sentence regarding the relationship between erythrocytes and oxylipins and inflammation.
We have added information on this now: blue mussels (source for n-3-fatty acids). Another important reason (which we discuss in our previous papers from the study) is that blue mussels are an environmental friendly seafood.
- And paragraph 2 (results) abbreviations appear LTB4, 7-HDoHE, 8-HETrE, 11-HDoHE, 90 15-oxoETE, 13-oxoODE, 20-HETE, and 8(9)-EET) - it is worth explaining them in the article.
We have now added explanations for these abbreviations.
- There are abbreviations throughout the article that have no explanation (e.g. fAA, fEPA, fDHA). Please re-read the text and explain all the abbreviations used.
We have now added explanations for these in row 111-112, instead of only explaining that free was abbreviated (f) in row 86.
- I understand that the full characteristics of the patients included in the study called MIRA were given in another article - reference number 20.
Yes, that is true.
Round 2
Reviewer 1 Report
No particular comments